# The Role of *Cdo1* in Ferroptosis and Apoptosis in Cancer

**DOI:** 10.3390/biomedicines12040918

**Published:** 2024-04-20

**Authors:** Xiaoyi Chen, Ansgar Poetsch

**Affiliations:** Queen Mary School, Nanchang University, Nanchang 330047, China; chenxiaoyi2003@126.com

**Keywords:** *Cdo1*, ferroptosis, apoptosis, cancer, cysteine

## Abstract

Cysteine dioxygenase type 1 (*Cdo1*) is a tumor suppressor gene. It regulates the metabolism of cysteine, thereby influencing the cellular antioxidative capacity. This function puts *Cdo1* in a prominent position to promote ferroptosis and apoptosis. *Cdo1* promotes ferroptosis mainly by decreasing the amounts of antioxidants, leading to autoperoxidation of the cell membrane through Fenton reaction. *Cdo1* promotes apoptosis mainly through the product of cysteine metabolism, taurine, and low level of antioxidants. Many cancers exhibit altered function of *Cdo1*, underscoring its crucial role in cancer cell survival. Genetic and epigenetic alterations have been found, with methylation of *Cdo1* promoter as the most common mutation. The fact that no cancer was found to be caused by altered *Cdo1* function alone indicates that the tumor suppressor role of *Cdo1* is mild. By compiling the current knowledge about apoptosis, ferroptosis, and the role of *Cdo1*, this review suggests possibilities for how the mild anticancer role of *Cdo1* could be harnessed in new cancer therapies. Here, developing drugs targeting *Cdo1* is considered meaningful in neoadjuvant therapies, for example, helping against the development of anti-cancer drug resistance in tumor cells.

## 1. Introduction

Cell death is an essential biological process found in all living organisms, serving various functions such as embryonic development, organ maintenance, aging, immune response coordination, and autoimmunity [1]. There exist alterations in cellular pathways either promoting or suppressing the pathways of cell death, which may result in cell immortality [1]. Cells proliferating uncontrollably and not dying are hallmarks of cancer [2]. In total, 18 million new cancer cases are diagnosed every year, and the most frequent cancer type is lung cancer followed by breast cancer, both of which amount to approximately 2.1 million cases [3]. One effector of cell death is cysteine dioxygenase type 1 (*Cdo1*), an enzyme that catalyzes the oxidation of cysteine. It plays roles in two different pathways of cell death, apoptosis and ferroptosis. In this review, we elaborate on how studying the anticancer role of *Cdo1* in apoptosis and ferroptosis will open new avenues to treat cancer.

### 1.1. Apoptosis and p53

Apoptosis is a programmed cell death that results in the orderly and efficient removal of cells. In normal cells, apoptosis occurs when there is severe damage or protein misfolding, including genetic mutation and imbalance in apoptotic factors. However, in cancer cells, the genetic variations lead to the misfunction of factors that inhibit cell division or induce apoptosis, enabling unlimited cell division and evasion of cell death [4]. Apoptosis occurs when the caspases are activated by pro-apoptotic factors [5]. Caspases refer to a group of enzymes of the cysteine protease family. The mitochondrion plays a critical role in apoptosis, as it releases pro-apoptotic factors. There are three pathways involved in apoptosis: the extrinsic pathway, intrinsic pathway, and intrinsic endoplasmic reticulum pathway [5]. p53 is a prominent tumor suppressor and nuclear transcription factor that induces expression of genes of the extrinsic and intrinsic pathways [6].

The extrinsic pathway is triggered once the death ligands are bound, for example, Fas ligand [7] to the Fas cell surface receptor, which is under transcriptional control by p53. Caspase 8 is involved in this pathway [7].

The intrinsic pathway, also known as the mitochondrial pathway, is triggered by irreparable genetic damage, hypoxia, severe oxidative stress, and high cellular Ca^2+^ levels, which are factors causing mitochondrial intrinsic damage [8]. These factors activate the pro-apoptotic factors Bax and Bac, which form pores in the mitochondrial membrane [5,9]. Bax and a subset of pro-survival genes of the Bcl2 family contain p53-binding elements [6]. Moreover, p53 exercises this function mainly by promoting the transcription of BH3-only proteins and the APAF-1 gene [10,11,12,13]. BH3-only proteins inhibit the Bcl2 family from blocking the formation of channels on the mitochondrial membrane and promote mitochondrial outer membrane permeabilization (MOMP), which in turn promotes apoptosis [5,14]. APAF-1 acts as a scaffolding protein to allow the activation of caspase-9 [15].

### 1.2. Ferroptosis

Ferroptosis is a type of regulated cell death marked by the accumulation of iron as well as iron-dependent lipid peroxides [16]. In some situations, ferroptosis also displays shedding and rounding up of cells, together with increased autophagosomes [17]. Iron promotes ferroptosis as a coactivator of lipoxygenases and a generator of reactive oxygen species (ROS) [17,18]. Without abundant antioxidants, ferroptosis is triggered as the radicals produced by the reaction of phospholipid hydroperoxide (PLOOH), a product of lipid peroxidation, and iron continue with the peroxidation of membrane lipids in a process called autoperoxidation [15], a positive feedback loop. In autoperoxidation, PLOOH reacts with both ferrous (Fe^2+^) and ferric (Fe^3+^) ions, resulting in the formation of the free radicals PLO^•^ and PLOO^•^, respectively [17], in the so-called Fenton reaction [19]. These free radicals then engage with polyunsaturated fatty acid–phospholipids (PUFA-PLs), promoting the continued production of PLOOH [17]. So far, three pathways with protecting roles against elevation of PLOOH have been studied.

The most well-known pathway protecting cells from ferroptosis involves GPX4 as the key element [17]. The cystine/glutamate antiporters transport cystine into cells. Cystine can be reduced to cysteine, an antioxidant [20]. Additionally, cysteine is essential for the production of glutathione (GSH), a powerful antioxidant [21]. Glutathione peroxidase 4 (GPX4) inhibits the Fenton reaction, by catalyzing the reduction of PLOOH to its alcoholic form PLOH, thereby protecting cells from lipid peroxidation and ferroptosis. In this process, GSH is involved as a cofactor of GPX4 (Figure 1).

There are two other newly discovered protecting pathways. Apoptosis-inducing factor mitochondria-associated 2 (AIFM2, also known as FSP1) and GTP cyclohydrolase-1 (GCH1) participate in these two pathways, respectively. Ferroptosis suppressing protein 1 (FSP1) inhibits lipid peroxidation and ferroptosis by converting ubiquinone (CoQ10)/semihydroquinone into ubiquinol, which can directly reduce lipid radicals, similar to GPX4 [22,23]. FSP1 catalyzes the protection process with the help of NADPH [22,23]. GCH1 is an enzyme that controls the rate of tetrahydrobiopterin (BH4) synthesis, which is essential for the production of neurotransmitters such as dopamine and nitric oxide (NO) [24]. BH4 acts as a cofactor for important enzymes involved in neurotransmitter and NO production [24]. Some types of phospholipids have polyunsaturated fatty acyl tails, such as phosphatidylserine and phosphatidylcholine, which are involved in regulating the fluidity and flexibility of the membrane [25]. Through GCH1-mediated BH4 production, ferroptosis is inhibited by selectively preventing the oxidation of lipids with two polyunsaturated fatty acyl tails [24] (Figure 1).

The pathway containing GPX4 is regulated by p53 [18]. p53 plays a vital role in inhibiting cell proliferation together with promoting apoptosis and ferroptosis [26]. In ferroptosis, p53 inhibits the expression of SLC755A, a subunit of the cystine/glutamate antiporter, which reduces the synthesis of the antiporter and inhibits the uptake of cysteine by the cells, thus depleting cells of antioxidants [27]. As result of this inhibition, cells develop ferroptosis [28]. Moreover, GPX4 is inhibited by a high level of Ca^2+^ [28]. Consequently, any ion channel eventually increasing intracellular Ca^2+^ facilitates ferroptosis. Piezo1, a mechanosensitive ion channel commonly produced in cancer cells, is essentially a calcium ion channel [28,29]. Therefore, cancer cells are more likely to undergo ferroptosis than ordinary cells in response to mechanical stimuli [29]. Chloride channels have also been found to have an effect on ferroptosis [30]. Transmembrane member 16A (TMEM16A) is a component of the Ca^2+^-activated chloride channel [30]. This channel allows Cl^−^ to enter the cell, and the abundance of anions in the cell encourages the cell to take up cations to maintain electric homeostasis, including Ca^2+^ [30]. The uptake of Ca^2+^ promotes ferroptosis [28].

Ferroptosis can be observed in both healthy tissues and tumors. Ferroptosis demonstrates toxic effects in healthy tissues. In the cardiovascular system, ferroptosis has been implicated in heart failure and atherosclerosis [31]. Similarly, in the nervous system, ferroptosis is observed in various neurological disorders, including neurodegenerative diseases, brain injuries, multiple sclerosis, aging, and neuroinflammation [32].

### 1.3. Cysteine Dioxygenase Type 1

Cysteine dioxygenase type 1 (*Cdo1*) is an enzyme that catalyzes the oxidation of cysteine; hence, it plays a pivotal role in cysteine metabolism by regulating the amount of cellular cysteine [20]. *Cdo1* catalyzes the conversion of cysteine to its sulfinic acid, which is further metabolized in the body [33]. Cysteine is essential for the production of glutathione (GSH), a vital cellular antioxidant [34]. Since the amount of available cysteine affects the sensitivity to oxidative stress in cells, the gene *Cdo1* plays a key role here [20]. Furthermore, oxidative stress is vital in inducing cell death, e.g., by apoptosis and ferroptosis. Induced cell death as an important direction in cancer chemotherapy [20], where *Cdo1* has been shown to increase the vulnerability of cells [35,36]. Moreover, and importantly, a decrease in the active *Cdo1* enzyme by mutation or downregulation can be seen in various types of cancers.

## 2. Ferroptosis and Apoptosis in Cancer

### 2.1. Ferroptosis in Cancer

#### 2.1.1. Two-Sided Role of Ferroptosis in Cancer

Ferroptosis, a type of programmed cell death, is initiated by cysteine deprivation and carried out through lipid peroxidation [18,37]. In cancer cells, metabolism is heightened, leading to elevated reactive oxygen species (ROS), and subsequently, increased oxidative stress [38]. Additionally, it has been demonstrated that cancer cells have a greater need for iron, which serves as a co-factor for numerous enzymes involved in ferroptosis and impacts lipid peroxidation [16]. Consequently, cancer cells are theoretically more vulnerable to ferroptosis, and indeed, this form of cell death has been frequently observed in various cancers, including fibroblastoma, lung cancer, osteosarcoma, kidney cancer, and prostate cancer [18].

Ferroptosis participates in eliminating unhealthy cancer cells, thereby promoting tumor growth to a limited extent [17]. However, in most cells within a tumor, ferroptosis predominantly hinders tumor growth or metastasis, highlighting its anticancer function [17].

#### 2.1.2. Inducing Ferroptosis to Combat Cancer

The human immune system is capable of inducing ferroptosis in tumor cells [39]; CD8^+^ T cells demonstrate their anticancer effects by releasing the cytokine IFN-γ, which triggers ferroptosis [39]. IFN-γ significantly reduces the expression of two subunits of the cystine/glutamate antiporter, SLC755A and SLC3A2, in tumor cells through activating the JAK-STAT signaling pathway [39]. This action decreases the antiporter and an inhibition of cellular uptake of cysteine, resulting in uncontrolled lipid peroxidation and ultimately ferroptosis [39]. Some tumor cells express PD-L1 on their surface, which binds to PD-1 on CD8^+^ T cells’ surface. As a result, these T cells fail to release cytokines, including IFN-γ, and do not exert cytotoxicity, allowing the bound tumor cell to evade the immune response [40]. Blocking this pivotal escape route has been pursued in immunotherapy; drugs that block the binding of PD-1 to PD-L1 allow CD8^+^ T cells not to be interfered with by PD-L1 on the tumor cell membrane, so that these CD8^+^ T cells can still exert cytotoxic effects to kill the cancer cell and release cytotoxic factors, including IFN-γ [41].

There are several drugs targeting ferroptosis directly. Among them, Erastin and RSL3 ((1S,3R)-RSL3) are representative [42]. Erastin has the ability to enhance the susceptibility of cancer cells to other anti-cancer drugs. Its mechanism of action involves inhibiting the expression of the subunit of the cysteine-glutamate antiporter SLC755A. This inhibition hinders the function of the antiporter, limiting the cells’ ability to uptake cysteine [43]. RSL3 binds directly to GPX4, inhibiting the action of GPX4, thus blocking the pathway that protects cancer cells from ferroptosis [44].

### 2.2. Apoptosis in Cancer

#### 2.2.1. Apoptosis and Carcinogenesis

Apoptosis presents tumor-suppressing roles in cancer [45]. Under physiological conditions, the mutated cancerous cells are detected by the immune system and killed [46]. By these means, apoptosis can prevent the formation of cancer through cell death [46]. However, one crucial change among the malignant genetic transformation of cells toward cancer is the ability to evade cell death, including apoptosis [46]. Therefore, when cells evade apoptosis, carcinogenesis develops. Cancer cells evade apoptosis mainly through three kinds of pathways: losing balance between pro-apoptotic and anti-apoptotic proteins, inhibiting caspases’ functions, as well as death receptor signaling [5]. For instance, p53 is involved in death signaling as well as inhibiting cell division, and its mutation induces cancer [47]. p53 mutation is associated with over 54% of cancer types, such as melanoma [47].

#### 2.2.2. Apoptosis in the Treatment of Cancer

Chemotherapeutic drugs induce apoptosis in cancer cells. For instance, Phikan088, a small molecule and carbazole derivative, has been proved to attach to mutated p53, thereby restoring the normal role of p53 in promoting apoptosis [48]. Therefore, developing drugs targeting molecules involved in apoptosis, such as BCL and p53, and regulating apoptosis, for example, *Cdo1*, is meaningful [49].

## 3. Ferroptosis and *Cdo1*

*Cdo1* promotes ferroptosis. Experiments inhibiting *Cdo1* production in mice indicate that *Cdo1* promotes ferroptosis by increasing the oxidative stress as well as inhibiting the production of GPX4 [36,50]. Upregulation of *Cdo1* in ferroptosis is thought to be regulated by c-Myb (c-Myb proto-oncogene transcription factor) by an unknown mechanism [36]. C-Myb encodes a transcription factor that directly interacts with the promoter of *Cdo1* [51]. C-Myb is thought to upregulate *Cdo1* in ferroptosis as cysteine enhances its DNA binding state, thereby upregulating the transcription of *Cdo1* [36]. In addition, both the expression of c-Myb and *Cdo1* are decreased in Erastin-induced ferroptosis in a dose-related fashion, meaning that the expressions of c-Myb and *Cdo1* are positively correlated [36]. Therefore, it is reasonable to infer that c-Myb upregulated *Cdo1* during ferroptosis [36].

High expression of *Cdo1* leads to significant ferroptosis, reducing the number of tumor cells in many cases, and plays a role in suppressing cancer rather than targeting healthy cancer cells [18]. *Cdo1* is part of the larger ferroptosis network that includes the cystine/glutamate antiporter and GPX4; therefore, the activity of each network member contributes to the activation or inhibition of ferroptosis (Figure 2). For example, the promotion of ferroptosis by *Cdo1* can be influenced by ion channels that increase Ca^2+^ influx, thereby inhibiting GPX4. If, in cancer, the expression of these ion channels is reduced or dysfunctional, intracellular Ca^2+^ can be reduced, and GPX4 function is preserved [29], which counteracts GSH depletion by *Cdo1* [18,20]. Moreover, dietary cystine availability should affect the concentration of cellular cysteine and thereby the amount that can be transformed by *Cdo1*. How nutrient availability is not only influenced by the diet, but also by the gut microbiome, is a current research question. Interestingly, it has been shown that traditional Chinese medicine (TCM) can elevate cysteine and methionine metabolism in the rat microbiome [52].

The off-target effects of *Cdo1* upregulation should be considered. *Cdo1* can promote ferroptosis based on the current study, but ferroptosis has negative aspects to the body other than cancer. For normal tissues, ferroptosis can easily cause diseases, for example, in the cardiovascular system, ferroptosis can lead to cardiomyopathy, myocardial infarction, and a series of cardiomyopathies, even heart failure [53,54]. In terms of metabolism, the off-target effect of *Cdo1* upregulation is still unclear and needs further study. In addition to drugs that specifically target ferroptosis, there are also a variety of agents that unexpectedly cause ferroptosis and are already in clinical use, such as cisplatin, sorafenib, and statin [55,56]. These drugs can lead to an off-target effect. Cisplatin promotes ferroptosis by inhibiting GPX4, and its off-target effect is mainly reflected in its nephrotoxicity [55,56,57]. Sorafenib induces ferroptosis by increasing intracellular iron levels and inhibiting the cystine/glutamate antiporter, and its off-target effect is mainly due to skin toxicity and causes diarrhea and arterial hypertension in patients [55,57,58]. Statin induces ferroptosis mainly by inhibiting two protective pathways containing GPX4 and FSP1, and its off-target effect is mainly to increase the Hemorrhagic Stroke Risk, as well as the myopathic effect [55,57,59].

## 4. Apoptosis and *Cdo1*

### 4.1. Detailed Tumor-Suppressing Role of Cdo1 in Cancer

Yang et al. found that in *Cdo1*-overexpressing breast cancer cells, the expression levels of tumor suppressors such as PTEN and BAX were increased, whereas the expression levels of proto-oncogenes such as PI3K and AKT, which promote cell division, were decreased [60]. In general, the gene expression analysis showed that *Cdo1*-overexpressing breast cancer cells had lower expression of metastatic and aggressiveness-related genes [60]. In cell culture, *Cdo1*-overexpressing cells divide much slower and form many fewer tribes than other tumor cells [60]. In breast cancer patients, those with *Cdo1* promoter hypermethylation showed a worse prognosis. This experiment illustrates the tumor-suppressing role of *Cdo1* both in terms of inhibiting cell division (PTEN, p53, PI3K, AKT) and promoting apoptosis (p53, BAX) [60].

### 4.2. Cdo1 Influences Lipid Peroxidation during Apoptosis

With *Cdo1* decreasing the cytosolic antioxidants cysteine and GSH, lipid peroxidation is more likely to happen [34]. It has been shown that lipid peroxidation is able to trigger and enhance apoptosis [18,61].

The products of lipid peroxidation affect the expression of elements involved in apoptotic signaling and cause DNA damage [62,63,64]. The inhibitor of kappa B kinase (IKK), a product of lipid peroxidation, is able to phosphorylate BCL proteins, enhancing apoptosis [65]. In addition, products of lipid peroxidation were found to form adducts with Jun N-terminal kinase (JNK), extracellular signal-regulated kinase (ERK), p38, and molecules activating mitogen-activated protein kinases (MAPKs), thereby activating these enzymes in apoptosis. Activated MAPKs are required for activating caspases, the executioners of apoptosis [66,67]. Furthermore, it is speculated that the lipid peroxidation product activates protein kinase C-delta (PKCδ) [68]. When PKCδ is cleaved by caspase-3, an activated catalytic fragment can be generated, amplifying apoptosis cascades [69].

### 4.3. Taurine, a Product of Cysteine Oxidation, Promotes p53 Activity

Taurine is the end product of cysteine oxidation catalyzed by *Cdo1* [20]. It was previously thought that taurine could only be produced in the liver, but it is now evident that taurine can also be produced in cells other than hepatocytes [70,71]. Taurine has an anti-tumor function, which promotes the production of p53, so that p53 can better perform its anti-oncogenic function [72,73].

Recent studies have found that elevated *Cdo1* expression can also enhance the tumor suppressor effects of p53 [35]. Given that the production of taurine requires the catalysis of *Cdo1*, it is appropriate to speculate that *Cdo1* enhances the tumor-suppressing effect of p53 by promoting the production of taurine. By increasing taurine expression of p53, *Cdo1* indirectly exerts a tumor-suppressing effect [20,35].

## 5. Mild Tumor-Suppressing Role of *Cdo1* in Cancer

Currently, a large number of studies have confirmed that increasing the expression level of *Cdo1* is able to inhibit cancer development by promoting cancer cell death, particularly by ferroptosis [60]. However, no study has found that silencing mutations in *Cdo1* alone can cause cancer.

In some cancer cells, *Cdo1* expression is promoted, but this promotion still fails to prevent carcinogenesis. This phenomenon implies that the anti-oncogenic effect of *Cdo1* is not strong [74]. Instead, *Cdo1* has a stronger effect on inhibiting malignancy than carcinogenesis [60]. The mild tumor-suppressing effect is illustrated for cells harboring mutation in the transcription factor Nrf2 [75]. Nrf2 promotes the role of *Cdo1* and facilitates taurine production. Both *Cdo1* and taurine have apoptosis-promoting properties, and *Cdo1* also has ferroptosis-promoting properties. Therefore, from the perspective of *Cdo1*, this mutation is favorable for the tumor-suppressing effect of *Cdo1* [75]. However, the overexpression of Nrf2 simultaneously promotes the uptake of cysteine, which is involved in the pathway that protects cells from ferroptosis [76]. This counteracts the anti-oncogenic effect of promoting *Cdo1* expression. The fact that cells remained cancerous in the presence of these two opposing effects suggests that the enhancement of the tumor-suppressing effect of *Cdo1* was not weaker than the promotion of cysteine uptake by Nrf2, and that cells were protected from ferroptosis [75].

## 6. *Cdo1* Alterations in Cancer Cells

Although *Cdo1* mutation is common in cancer (Figure 3, downloaded from cBioportal (April 2024)) [77], it must be clarified that mutations in the *Cdo1* gene are not driver mutations of cancer [77]. Furthermore, it is worth mentioning that due to the role of *Cdo1* in enhancing apoptosis and cell cycle arrest, but not inducing apoptosis or cell cycle arrest, it is suggested that *Cdo1* mutation cannot be the driver mutation leading to cancer [78].

### 6.1. Genetic Mutations and Structural Variations

Observed genetic mutations comprise missense mutation, splicing mutation, and truncating mutation (Figure 4, downloaded from cBioportal (April 2024)) [78]. Missense mutation is the most common type, accounting for about 82% of all genetic alterations of *Cdo1* (Figure 4) [78]. Splicing mutations and missense mutations have similar outcomes: the production of non-functional *Cdo1* [78]. Structural variation is even less common than genetic mutation [78]. Moreover, *Cdo1* mutation can be seen in every cancer type, but only in a low percentage [78], with the highest percentage in lung cancer being still less than 10% (Figure 4). Cancer types in which *Cdo1* mutation and structural variation are relatively common are melanoma, breast cancer, and pancreatic cancer [77]. Since *Cdo1* mutation is not the driver mutation leading to tumorigenesis, it is understandable that *Cdo1* genetic alterations occur at low frequency. Figure 3 demonstrates that missense mutations have occurred at nearly every position of the *Cdo1* gene. The most common splice mutation usually occurs at the beginning and the end of exon 2. There exist truncations in the middle of exon 1 and the middle of exon 2.

### 6.2. Epigenetic Silencing of Cdo1

Epigenetic silencing is cancer-specific, and the most obvious form is methylation of the *Cdo1* promoter [78]; generally, methylation of a gene’s promoter inhibits its transcription [79]. Methylation of the *Cdo1* promoter can be observed in various cancers, including colorectal cancer, breast cancer, and gastric cancer [80,81,82]. Owing to its frequent occurrence in cancers, *Cdo1* methylation has become a biomarker for various types of cancers [83], and blood testing for the methylated *Cdo1* promoter offers useful information in detecting cancer in patients [83]. It is reasonable to assume that *Cdo1* expression can be regulated through histone modification [84]. However, the effect of histone modification on *Cdo1* expression is not clear. Therefore, this section will focus on the mechanism of DNA methylation.

Currently, the mechanism of methylation of the *Cdo1* promoter is not fully understood. There are two theories. ① The methylation of the targeted CpG sequence by DNA methyltransferase (DNMT) is enhanced [85]; DNA methyltransferase adds methyl groups to the nucleotides, especially the CpG nucleotide [86]. ② This specific gene methylation is just a consequence of cell proliferation, especially in acute mononuclear leukemia [87]. Two possible mechanisms supporting the first theory are introduced (Figure 5).

The first theory suggests that a methylated tumor suppressor gene makes the cell cancerous [88]. The methylation of the *Cdo1* promoter is induced by Chd4/NuRD chromatin remodeling factors in tumor cells [88,89]. In normal cells, the Chd4/NuRD chromatin remodeling factors appear to cue the DNMT site of action only after DNA damage to *Cdo1* [90]. Such methylation in a tumor suppressor gene can directly make the cell cancerous, leaving the tumor suppressor gene promotor methylated and non-functional [88]. The exact mechanism of this phenomenon has not been investigated [88]. However, regarding the mild tumor-suppressing effect of *Cdo1*, it could be possible that this epigenetic methylation is generated after carcinogenesis because cancer induced by *Cdo1* silencing has not been found.

Another possible mechanism supporting the first theory is the ten-eleven translocation (TET)-mediated pathway dysregulation [88]. TET-dependent 5-hydroxymethylation is crucial in preserving the unmethylated state of normal cells’ CpG islands, thus exhibiting the opposite function to DNMT [90,91]. The most recent study found that hypermethylation is relevant to a bivalent promoter [92]. This study illustrated that some promoters showed partial methylation of the CpG island edges, resulting in the formation of smaller, unmethylated CpG islands [88]. The adjacent borders of these smaller unmethylated CpG islands are targeted by DNMT3A and TET. 5hmC (5-hydroxymethylcytosine) and DNA methylation inhibits binding, maintaining the length of the island [93,94]. The protein recruiting DNMT4 and TET in this process is H3K4me1 [92]. This protein marks the TET-mediated 5hmC at the borders of the unmethylated CpG islands [95]. In cancer cells, the activity of TET is lost, and the 5hmC level is low, indicating that there is no competition with DNMT and the binding of DNMT is allowed. This mechanism is seen in acute myeloid leukemia (AML), where CpG island methylation contributes to the upregulated proliferation of cancer cells in AML [95]. Considering the mechanism of epigenetic silencing of tumor-suppressing genes in cancer, targeting DNMTs in cancer therapy is supposed to be a powerful direction in chemotherapy [88].

Since the expression of *Cdo1* is inhibited by methylation, its tumor-suppressing effects, including the promotion of apoptosis and ferroptosis, are limited [60]. Therefore, epigenetic methylation has an important inhibitory effect on the function of *Cdo1*. In cancer cells, methylation is the most common variant of *Cdo1*, and although it is not sufficient to make cells cancerous, the presence of this variant can exacerbate the progression of cancer.

## 7. Conclusions and Outlook

*Cdo1* is considered a tumor suppressor as it enhances ferroptosis and apoptosis in cancer cells. Mechanistically, *Cdo1* enhances ferroptosis by promoting cysteine starvation, thereby also reducing GSH levels [17]. GSH is the cofactor of GPX4, the enzyme protecting a cell from autoperoxidation of membranes [17] (Figure 6). In conclusion, *Cdo1* conducts its tumor-suppressing role by reducing the antioxidants protecting cells from ferroptosis [20]. Moreover, *Cdo1* promotes apoptosis by aggravating lipid peroxidation as well as upregulating the activity of p53 [60,61] (Figure 6).

Although the tumor-suppressing effect of *Cdo1* is mild, it plays important roles in assisting anticancer agents in overcoming drug resistance. For instance, *Cdo1* plays a significant role in mediating Erastin-induced ferroptosis in gastric cancer cells [36]. When the *Cdo1* gene is silenced, ferroptosis does not occur even though Erastin is applied [36]. Furthermore, epigenetic silencing of *Cdo1* contributes to resistance against ROS-generating chemotherapeutic drugs, including anthracycline [96]. Therefore, upregulating *Cdo1* in cancer cells is likely to assist the effect of major chemotherapeutic agents.

In conclusion, *Cdo1* conducts the tumor-suppressing role in cancer cells by contributing to ferroptosis as well as enhancing the anti-tumor effects of p53, thereby mediating apoptosis and cell cycle arrest. Furthermore, p53 enhances ferroptosis (Figure 6). Therefore, *Cdo1* is a useful cancer drug target. Based on the current literature, combined anticancer therapy ensuring high *Cdo1* activity should be a powerful approach to ensure and enhance the effect of anti-cancer drugs and to impede the development of drug resistance in cancer cells.

## Figures and Tables

**Figure 1 biomedicines-12-00918-f001:**
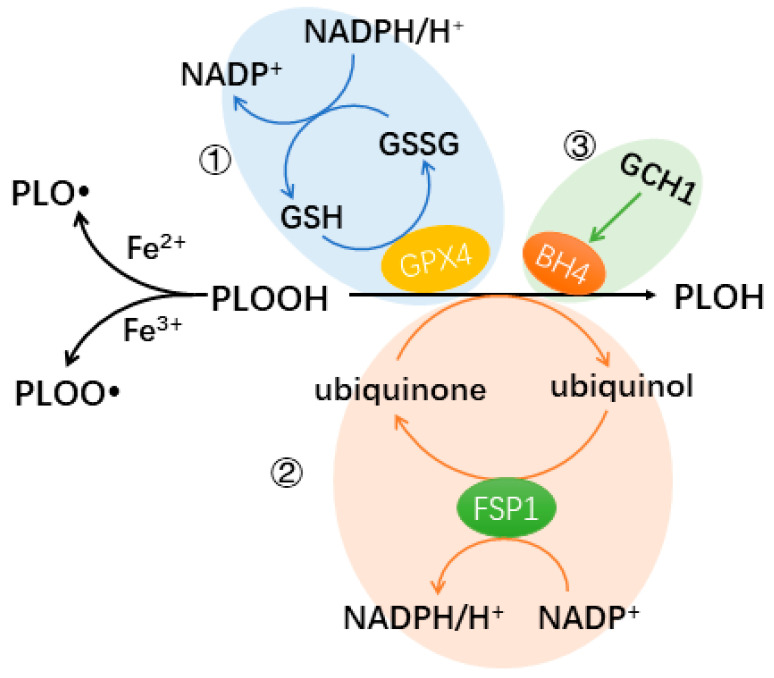
Known pathways protecting cells from ferroptosis. The 3 known pathways involve GPX4 (①), FSP1 (②), and BH4 generated from GCH1 (③), respectively. The 3 pathways have the same effect: protecting the cells from further lipid peroxidation by reducing PLOOH levels. Iron [II/III] reacts with PLOOH via the Fenton reaction to produce free radicals, which can induce the peroxidation of polyunsaturated fatty acids (PUFAs), promoting the loss of membrane integrity and cell death. (GSH: glutathione, GSSG: oxidized glutathione, GPX4: glutathione peroxidase 4, PLOOH: phospholipid hydroperoxide, PLOH: the corresponding alcohol of PLOOH, FSP1: ferroptosis suppressing protein 1, GCH1: GTP cyclohydrolase-1, BH4: tetrahydrobiopterin).

**Figure 2 biomedicines-12-00918-f002:**
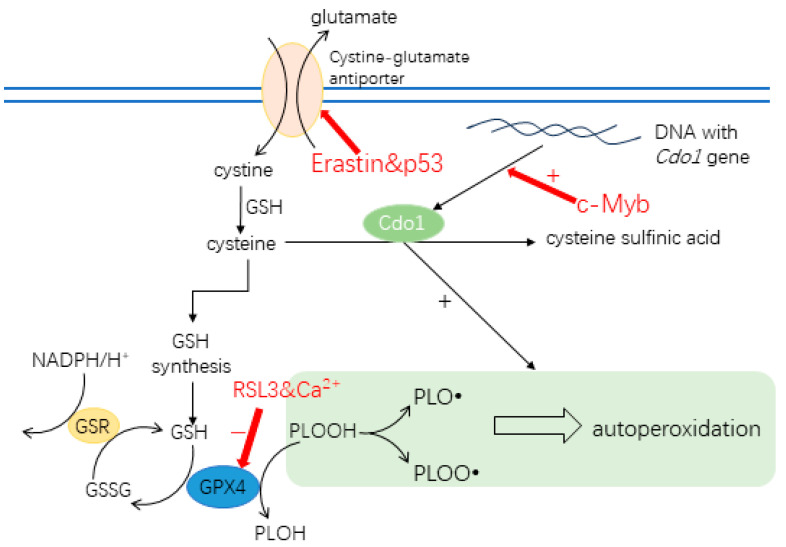
Function of *Cdo1* in ferroptosis and targets of Erastin and RSL3 for inducing ferroptosis. Erastin inhibits the cystine/glutamate antiporter by binding to the subunit SLC755A, leading to cysteine deprivation by suppressing cystine uptake and triggering ferroptosis. Similarly, p53 inhibits the synthesis of SLC755A, the subunit of the cystine-glutamate antiporter, suppressing the uptake of cysteine. RSL3 directly binds to GPX4, inhibiting its function. Calcium ions inhibit GPX4 function, too. C-Myb is supposed to regulate *Cdo1* expression in ferroptosis by an unknown mechanism. Enzymatic conversion by *Cdo1* lowers the cysteine concentration, thereby depleting the pool available for the formation of GSH. Consequently, the autoperoxidation of lipids by the Fenton reaction and PLOOH cannot be inhibited, leading to ferroptosis. (GPX4: glutathione peroxidase 4, PLOOH: phospholipid hydroperoxide, PLOH: the corresponding alcohol of PLOOH, *Cdo1*: cysteine dioxygenase type 1, GSR: glutathione-disulfide reductase, GSSG: oxidized glutathione, GSH: glutathione, NADPH: triphosphopyridine nucleotide, c-Myb: c-Myb proto-oncogene transcription factor, RSL3: Ras-selective lethal 3).

**Figure 3 biomedicines-12-00918-f003:**
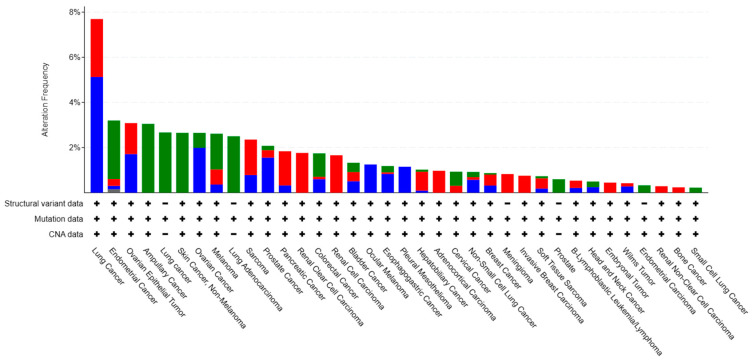
Percentage of *Cdo1* mutations and structural variants in cancers. Overview of cancer types that contain *Cdo1* mutations and structural variants. Copy number alterations means the change in the number of copies of a particular region of DNA in the genome, involving either a gain or a loss of copies of a specific DNA segment. *Cdo1* mutations appear in nearly every kind of cancer; the figure displays only cancers with a relatively high percentage of *Cdo1* genetic variation. Source: CBioportal (2024). Cancers with *Cdo1* mutation [Pancancer database] (https://www.cbioportal.org/).

**Figure 4 biomedicines-12-00918-f004:**
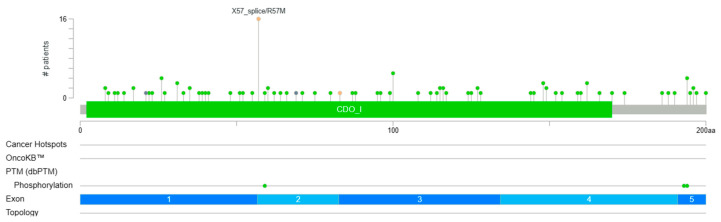
Positions of genetic alterations on *Cdo1*. This lollipop figure shows the mutated amino acid positions on the *Cdo1* protein and the frequency of observation in patients. Source: CBioportal (2024). Types of *Cdo1* mutations in cancers [Pancancer database] (https://www.cbioportal.org/).

**Figure 5 biomedicines-12-00918-f005:**
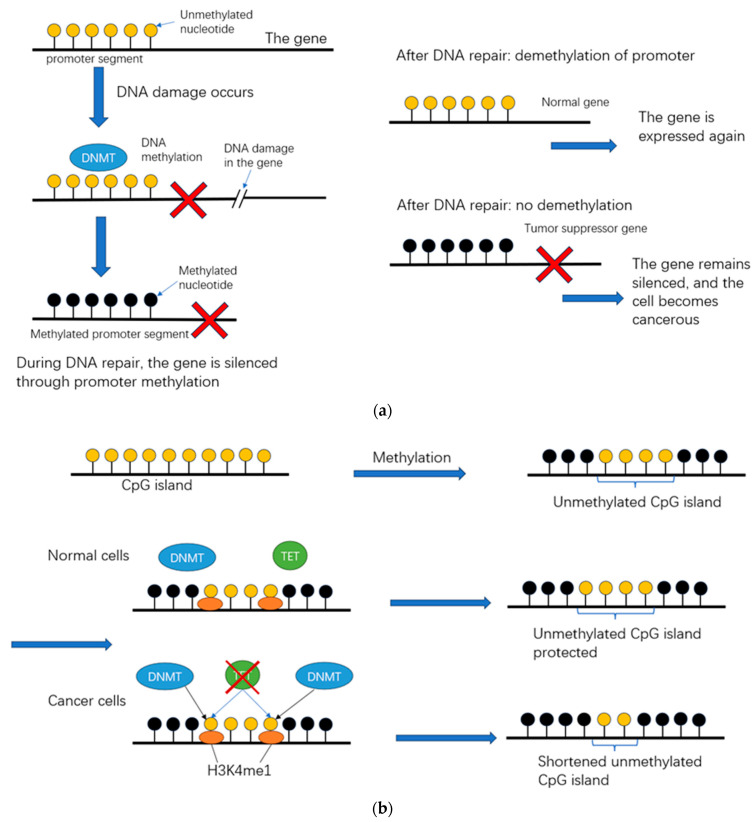
Theories explaining the methylation of the *Cdo1* promoter in cancer. (**a**) When DNA damage occurs to a tumor suppressor gene, the methylation of the promoter occurs to inhibit the expression of the damaged gene while repairing. When the damaged gene is an anticancer gene, this inhibition can be related to carcinogenesis. In this case, by an unknown mechanism, the methylation of this promoter is not re-exposed in cancer cells for removal. (**b**) Alternative mechanism for *Cdo1* methylation in cancer cells. Cancer cells contain low levels of 5hmC, an element that inhibits the binding of TET and DNMT to the adjacent shore of the unmethylated CpG island. The binding is guided by H3K4me1. In normal cells, the binding of DNMT and TET is largely inhibited by the methylated DNA and 5hmC. Therefore, the unmethylated CpG island is maintained. However, in cancer cells, the 5hmC level is low, H3K4me1 appears at the border directing the binding of DNMT, and TET activity is lost. Only DNMT works on the shore of the island, and the shore is methylated, shortening the island. With a larger portion of the promoter methylated, the expression of *Cdo1* is downregulated.

**Figure 6 biomedicines-12-00918-f006:**
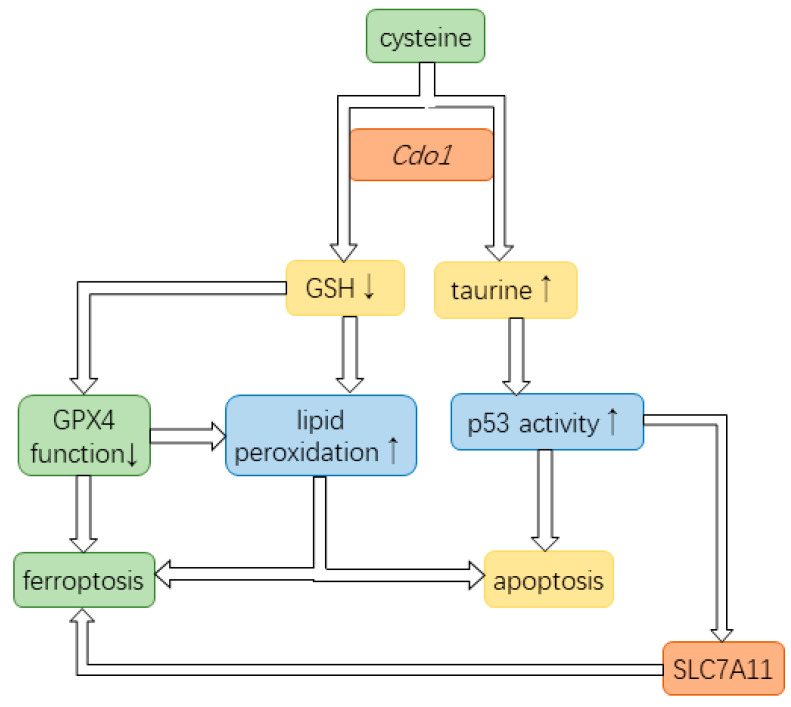
*Cdo1* partakes in pathways of ferroptosis and apoptosis. *Cdo1* is involved in converting cysteine to taurine, thereby decreasing the cysteine pool for the production of GSH. GSH is a required cofactor of GPX4 that prevents lipid peroxidation. The reduced function of GPX4 allows lipid peroxidation on the cell membrane, ultimately leading to cell death. Upregulated taurine can increase the level of p53, a vital tumor suppressor. Additionally, p53 can induce cell cycle arrest. Lipid peroxidation increases the likelihood of apoptosis and ferroptosis. p53 contributes to ferroptosis by inhibiting the production of the SLC755A, a subunit in the cystine-glutamate antiporter that uptakes cystine.

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
