# Peer review of "The Role of Cdo1 in Ferroptosis and Apoptosis in Cancer"

_biomedicines, 2024, doi:10.3390/biomedicines12040918_

Round 1

Reviewer 1 Report

Comments and Suggestions for Authors

1.Cdo1 has a weak inhibitory effect in cancer, so isn't this paper meaningless?

2.The importance of traditional medicine is increasing in ferroptosis and apoptosis. If there is a study on Cdo1 relevance in traditional medical research, please mention it in discussion.

3.The importance of ion channels in cancer research is also mentioned in many papers. Please mention the relationship between Cdo1 and the ion channel.

Author Response

1.Cdo1 has a weak inhibitory effect in cancer, so isn't this paper meaningless?

As stated in the manuscript, Cdo1 is a mild tumor suppressor. It is not very important in preventing cancer formation, but development of malignancy - which we consider as important. We elaborated this in section 5. Indeed, as for many anti-cancer genes, mutations have been observed that can overcome the effect of Cdo1, and we mentioned mutation in Nrf2 gene. 

In our opinion, in respect to inhibitory effect of Cdo1 this paper is meangingful in at least two aspects, first Cdo1 is a relevant tumor suppressor given its previously observed mutations and structural variants in cancer (in particular lung cancer). Second, inducing ferroptosis and apoptosis in cancer cells is a highly relevant treatment approach, and we demonstrate that Cdo1 is a key player in both processes.

2.The importance of traditional medicine is increasing in ferroptosis and apoptosis. If there is a study on Cdo1 relevance in traditional medical research, please mention it in discussion.

We did not find a study that directly investigates the effect of TCM on Cdo1 function. However, we found a work that demonstrates elevation of cysteine and methionine metabolism in gut microbiome by TCM, suggesting an indirect effect on Cdo1 function due to likely changed cellular cystine availability: "Interestingly, it has been shown that traditional Chinese medicine (TCM) can elevate cysteine and methionine metabolism in the rat microbiome [52]."

3.The importance of ion channels in cancer research is also mentioned in many papers. Please mention the relationship between Cdo1 and the ion channel.

Thank you very much for your remark, we searched literature again and found a possible link between Ca2+ uptake and function of GPX4. We updated Figure 2 and manuscript text accordingly: "For example, the promotion of ferroptosis by Cdo1 can be influenced by ion channels that increase Ca2+ influx thereby inhibiting GPX4. If in cancer the expression of these ion channels is reduced or dysfunctional, intracellular Ca2+ can be reduced, and GPX4 function is preserved [29], which counteracts GSH depletion by Cdo1. "

Reviewer 2 Report

Comments and Suggestions for Authors

The manuscript provides a detailed account on the role of Cdo1 in ferroptosis and apoptosis in cancer. The manuscript is well structured, and information is presented comprehensively. But there are few suggestions which can help readers to grasp the topic more efficiently.

1.        Since that Cdo1 is expressed in non-cancerous tissues as well, targeting it as a therapeutic target may have unexpected effects on normal cellular processes. Author could include studies which have shown any off-target effects and toxicity of Cdo1 inhibition.

2.        Author should expand the section epigenetic silencing of Cdo1 as this is the pivotal mechanism of Cdo1 dysregulation in cancers.

3.        Almost all studies included in the manuscript are invitro based , consider including the in vivo studies in which researchers have shown how invitro findings translate in in vivo settings.

4.        Consider reorganizing or condensing some sections to improve the flow and clarity of the manuscript.

Author Response

1. Since that Cdo1 is expressed in non-cancerous tissues as well, targeting it as a therapeutic target may have unexpected effects on normal cellular processes. Author could include studies which have shown any off-target effects and toxicity of Cdo1 inhibition.

We have consulted the literature about known off-target effects for drugs against Cdo1 and ferroptosis. These are now part of the manuscript:

"The off-target effects of Cdo1 upregulation should be considered. Cdo1 can promote ferroptosis from the current study, but ferroptosis has negative aspects to the body other than cancer. For normal tissues, ferroptosis can easily cause diseases, for example, in the cardiovascular system, ferroptosis can lead to cardiomyopathy, myocardial infarction and a series of cardiomyopathies, even heart failure [53,54]. In terms of metabolism, the off-target effect of Cdo1 upregulation is still unclear and needs further study. In addition to drugs that specifically target ferroptosis, there are also a variety of agents that unexpectedly cause ferroptosis and are already in clinical use, such as cisplatin, sorafenib, statin [55,56]. These drugs can lead to off-target effect. Cisplatin promotes ferroptosis by inhibiting GPX4, and its off-target effect is mainly reflected in its nephrotoxicity [55-57]. Sorafenib induces ferroptosis by increasing intracellular iron levels and inhibiting the cystine/glutamate antiporter, and its off-target effect is mainly due to skin toxicity and causes diarrhea and arterial hypertension in patients [55,57,58]. Statin induces ferroptosis mainly by inhibiting two protective pathways containing GPX4 and FSP1, and its off-target effect is mainly to increase the Hemorrhagic Stroke Risk, as well as the myopathic effect [55,57,59]."

2.        Author should expand the section epigenetic silencing of Cdo1 as this is the pivotal mechanism of Cdo1 dysregulation in cancers.

We agree about the importance of this aspect, however, could not find additional information about epigenetic mechanisms and Cdo1. We mentioned the reasonable role of histone modifications - maybe future research will provide a better understanding here: "It is reasonable to assume that Cdo1 expression can be regulated through histone modification [87]. However, the effect of histone modification on Cdo1 expression is not clear. Therefore, this section will focus on the mechanism of DNA methylation."

3. Almost all studies included in the manuscript are invitro based , consider including the in vivo studies in which researchers have shown how invitro findings translate in in vivo settings.

We agree that in vivo studies are necessary to validate in vitro data. Unfortunately, there is a lack of research, we could find only two examples, now mentioned in the revision:

"Cdo1 promotes ferroptosis. Experiment inhibiting Cdo1 production in mice indicates that Cdo1 promotes ferroptosis by increasing the oxidative stress as well as inhibiting the production of GPX4 [35,50]. "

4.        Consider reorganizing or condensing some sections to improve the flow and clarity of the manuscript.

The sections about apoptosis and ferroptosis have been condensed, since there exists a wealth of literature for these topics. Additionally, several small revisions  have been made.